# Sex hormone profiles and associated factors among adult tuberculosis patients at Gondar town, northwest Ethiopia: A comparative cross-sectional study

Eshet Gebrie[1]*, Habtamu Wondifraw Baynes[1], Berihun Agegn Mengistie[2], Temesgen Kassie[3], Zeleke Kassahun[3], Abebe Birhanu[4], Amanuale Zayede[4], Elias Chane[1]

1 Department of Clinical Chemistry, School of Biomedical and Laboratory Sciences, College of Medicine and Health Sciences, University of Gondar, Gondar, Ethiopia, 2 Department of General Midwifery, School of Midwifery, College of Medicine and Health Sciences, University of Gondar, Gondar, Ethiopia, 3 Department of Clinical Chemistry, University of Gondar Comprehensive and Referral Hospital, Gondar, Ethiopia, 4 Department of Medical Microbiology, School of Biomedical and Laboratory Sciences, College of Medicine and Health Sciences, University of Gondar, Gondar, Ethiopia

* eshetgebrie@gmail.com

## Abstract

### Background

Tuberculosis, caused by Mycobacterium tuberculosis, is the second leading cause of death from infectious diseases worldwide. Tuberculosis is associated with alterations in sex hormone levels, particularly testosterone, estradiol, and progesterone. However, previous studies have reported conflicting results, with some showing increased or decreased levels in tuberculosis positive patients, while others found no significant differences. This study aims to assess and compare sex hormone profiles among adult tuberculosis-positive patients and tuberculosis-negative individuals and to identify associated factors.

### Method

A comparative cross-sectional study was conducted from June 15 to August 20, 2024, among 300 eligible adult tuberculosis-positive patients and age-matched tuberculosis-negative individuals in five selected health institutions in Gondar town. Participants were recruited using a simple random sampling technique, and socio-demographic, clinical, anthropometric, and behavioral data were collected through a structured questionnaire. Five milliliters of venous blood were used to determine hormone levels using the Beckman Coulter DXI 800 chemistry hormonal analyzer. Hypogonadism was defined by sex-specific hormones and categorized as primary, secondary, and subclinical. The data were analyzed using SPSS version 25.0. Descriptive statistics, independent t-tests, one-way ANOVA, Mann-Whitney U,

**Data availability statement:** All relevant data are within the paper and its Supporting Information files.

**Funding:** The author(s) received no specific funding for this work.

**Competing interests:** The authors have declared that no competing interests exist.

**Abbreviations:** BMI: Body Mass Index; DM: Diabetes Mellitus; DBP: Diastolic Blood Pressure; EPTB: Extrapulmonary Tuberculosis; FSH: Follicle Stimulating Hormone; GnRH: Gonadotropin Releasing Hormone; HG: Hypogonadism; HPG: Hypothalamus–Pituitary–Gonadal; IL: Interleukin; LH: Luteinizing Hormone; MTB: Mycobacterium Tuberculosis; PTB: Pulmonary Tuberculosis; SBP: Systolic Blood Pressure; TB: Tuberculosis; TGFβ: Tumor Growth Factor Beta; TNFα: Tumor Necrosis Factor Alpha; UGRCSH: University of Gondar Referral Compressive Specialized Hospital; WHO: World Health Organization.

Kruskal-Wallis H test, and bivariable and multivariable statistical models were used. A p-value $< 0.05$ with a 95% CI was considered statistically significant.

## Result

Male tuberculosis-positive patients showed significantly higher estradiol, luteinizing hormone, and FSH ($p < 0.001$), but lower testosterone ($p < 0.001$). Newly diagnosed tuberculosis-positive patients had significantly lower progesterone levels ($p < 0.005$). Female tuberculosis-positive patients showed significantly lower testosterone and progesterone but higher follicle-stimulating hormone levels compared to tuberculosis-negative individuals ($P < 0.001$). Estradiol and luteinizing hormone levels did not differ significantly in female tuberculosis-positive patients. However, newly diagnosed tuberculosis-positive patients had significantly higher median estradiol levels ($p < 0.001$). The overall prevalence of hypogonadism was 30.3% (95% CI (25.2–35.9%)), sex [AOR = 11.36, 95% CI (3.6, 36.17)] ($p < 0.001$), dietary diversity (participants with lower diversity, including those with no dietary diversity [AOR = 8.98, 95% CI (2.37, 33.99)] ($p = 0.001$), those with sometimes [AOR = 9.2, 95% CI (2.77, 30.62)] ($p < 0.001$) and a usual dietary diversity [AOR = 3.24, (1.04, 10.06) ($p = 0.042$), and cortisol [AOR = 4.01, 95% CI (1.7, 9.5)] ($p = 0.002$) levels were significant determinants of hypogonadism in tuberculosis patients.

## Conclusion

Male tuberculosis-positive patients showed significantly higher mean estradiol, luteinizing hormone, and follicle-stimulating hormone but lower testosterone levels, while female tuberculosis-positive patients had significantly lower testosterone and progesterone and higher follicle-stimulating hormone levels. There was a higher prevalence of hypogonadism, particularly in males. Early diagnosis of these hormonal changes is crucial in tuberculosis-positive patients to prevent complications such as impaired gonadal function related to hormonal disturbances.

## Introduction

Tuberculosis (TB) is a chronic airborne infectious disease caused by the bacillus *Mycobacterium tuberculosis* (MTB). Tuberculosis usually affects lungs, causing pulmonary tuberculosis (PTB); however, it can also involve almost all other organs except nails, teeth, and hair other than the lungs and is referred to as extrapulmonary tuberculosis (EPTB) [1,2]. Tuberculosis has evolved into the single infectious disease causing morbidity and mortality among millions across the world. According to a recent global TB estimate by the World Health Organization (WHO), there were 7.5 million new cases and 1.3 million deaths globally in 2022. After Coronavirus disease 19 (COVID-19), TB continued to be the second most common infectious agent-related cause of death worldwide in 2022 [3]. Each year, more than 10 million people still contract TB. Most cases are in low- and middle-income countries (LMIC), including Ethiopia [3,4].

Infections, including TB, can significantly alter hormonal levels [5]. The hypothalamus-pituitary-gonadal (HPG) axis is the critical neuroendocrine system that regulates the functions of the testes in males and the ovaries in females, controlling the production and release of sex hormones. Sex steroid hormones, namely progesterone, estrogen, and testosterone, are produced by gonads and the adrenal gland. In addition to their roles in reproduction and sexual differentiation, sex hormones such as estradiol, progesterone, and testosterone also influence the immune system [6].

There is ample evidence to support the well-known bidirectional contact between the immune system and the neuroendocrine system. In response to TB infection, cytokines like interleukin one (IL-1), interleukin six (IL-6), and tumor necrosis factor-alpha (TNFα), produced by immune cells, can activate the (HPG) axis [7]. Immune cells have sex hormone receptors, and during the early host contacts with MTB, their steroid signaling pathways may affect functional responses of immune cells and also affect the course of TB infection processes [8].

Males and females are rarely equally affected by infectious disease, and TB is not the same for both genders [9]. Male adults have a 1.7-fold higher risk of developing active TB than adult females do globally [10,11]. This disparity of sex was likely attributed to a combination of sociocultural roles, behavioral factors, and biological differences that may make men more susceptible to TB [11]. But this gender difference does not apply to children and young adolescents [12,13] and could arise after sexual maturation (after puberty). The gender bias in tuberculosis suggests that sex hormones play a part in the immunoregulation of the infection and affect a person's susceptibility and response to infections and are linked to poor patient outcomes and higher fatality rates [14,15].

Tuberculosis infection has been shown to significantly alter serum sex hormone levels, largely through immune-mediated mechanisms [16,17]. In TB patients, decreased testosterone is observed [16,18], likely due to impaired Leydig cell steroidogenesis resulting from cytokine-mediated inhibition of luteinizing hormone (LH) action. Increased pro-inflammatory cytokines (e.g., Tumor Growth Factor Beta (TGF-β)) within the testes contribute to this testosterone reduction [17,19]. In advanced active TB, increased estrogen synthesis occurs through the aromatization of testosterone [16,20], a process facilitated by the aromatase activity of these same pro-inflammatory cytokines [21]. These cytokines also stimulate the hypothalamus to release gonadotropin-releasing hormone (GnRH), leading to increased pituitary follicle-stimulating hormone (FSH) and LH production and potentially increased estradiol by gonads [22,23]. However, EPTB can cause gonadal abscess, potentially reducing estradiol levels [24–26]. In TB patients, progesterone levels are significantly decreased [27]; this may be due to elevated pro-inflammatory cytokines produced in response to MTB antigen that suppress progesterone secretion by the ovaries [28]. Tuberculosis impact on sex hormones is complex, involving both direct [24] and indirect effects mediated by pro-inflammatory cytokines [17,19].

Recently there has been a rise in global TB cases, deaths, and the spread of highly drug-resistant strains, providing alarming signals that other strategies will be needed to stop this endemic disease [29]. Global control of TB can only be achieved through the concerted effort in the development of effective vaccines and improved diagnostics, as well as novel and shortened therapeutic regimens. The findings of this study will contribute to the identification of additional biomarkers that add to the existing body of knowledge towards developing potential host markers for diagnostics, prognostics, or vaccine development initiatives. Therefore, this study was aimed to assess sex hormone profile and associated factors in adult TB-positive patients and compare with apparently healthy TB-negative comparison group.

## Materials and methods

### Study design, period, and area

A multi-center institution-based comparative cross-sectional study was conducted from June 15 to August 20, 2024. Five selected governmental health institutions in Gondar Town (University of Gondar Referral Comprehensive Specialized Hospital (UGRCSH), Marakie, Azezo, Poli, and Mitwab health center) were involved in this study. Gondar town is located in

the Central Gondar Zone of the Amhara regional state, Ethiopia, approximately 627 kilometers northwest of Addis Ababa, the capital city of Ethiopia, and about 180 kilometers from Bahir Dar, the capital city of the Amhara regional state, with a total population of 412,739 as per the year 2023 [30]. Gondar city has eight health centers, 14 health posts, 32 private clinics, and one comprehensive specialized hospital [31].

### Source and study population

**Source population. Tuberculosis-positive patients:** All diagnosed TB-positive patients aged 18 years and above who visited or were admitted to health institutions in Gondar town.
**Tuberculosis-negative comparison group:** All apparently healthy TB-negative adults aged 18 and older who visited Gondar town health institutions for another purpose were the source population for the TB-negative comparison group. Tuberculosis-positive patients and TB-negative comparison group were matched by age.

### Study population

**Tuberculosis-positive patients:** All diagnosed TB-positive patients aged 18 and above who were visiting or admitted and fulfilling the inclusion criteria were the study population for Tuberculosis-positive patients.
**Tuberculosis-negative comparison group:** All apparently healthy TB-negative adults aged 18 and older who were available in Gondar town health institutions during the study period and gave informed consent were the study population for TB-negative comparison group.

### Eligibility criteria

**Inclusion criteria. Tuberculosis-positive patients:** All adult patients aged 18 and above diagnosed with TB (newly diagnosed individuals and individuals on treatment and able to provide informed consent during the study period were included in the study.
**Tuberculosis-negative comparison group:** All apparently healthy TB-negative adults aged 18 and older who were willing to participate in the study were included in the TB-negative comparison group.

### Exclusion criteria

**Tuberculosis-positive patients** (adult TB patients aged 18 and above) were excluded if they:
  Were pregnant women or lactating women.
  Were receiving medications affecting sex hormone metabolism (i.e., dopamine, opioids, glucocorticoids, beta-blocker, and exogenous estrogen therapy) during the study period.
  Had coexisting conditions such as Human Immunodeficiency Virus/Acquired Immunodeficiency Syndrome (HIV/AIDS), Diabetes Mellitus (DM), pre-existing hypertension, liver disease, kidney disease, Cushing syndrome, and hyperthyroidism (based on medical record review).
  **Tuberculosis-negative comparison group** (apparently healthy TB-negative adults aged 18 and above) were excluded if they:
  Were pregnant women or lactating women.
  Were receiving medications affecting sex hormone metabolism (i.e., dopamine, opioids, glucocorticoids, beta-blockers, and exogenous estrogen therapy) during the study period.
  Had known HIV/AIDS, DM, pre-existing hypertension, liver disease, kidney disease, Cushing syndrome, and hyperthyroidism (based on self-reporting).
  Had close contact with TB-positive patients (such as laboratory workers in TB units, nurses, or family caregivers of TB-positive patients).

## Study variables

**Dependent variable.** Sex hormone profile (testosterone, estradiol, progesterone, LH, and FSH) and HG

**Independent variable.** Sociodemographic variables (age, sex, marital status, residency, educational status, occupation status and monthly income), behavioral factors (tobacco smoking, alcohol drinking, diet, and physical activity), and anthropometric, biochemical, and clinical variables (body mass index (BMI), blood pressure, cortisol level, TB classification, history of TB, treatment status, duration of treatment, medication other than anti-TB drugs, family history of medical disease, presence of vomiting, nausea, and loss of appetite, and comorbidities (malaria, giardiasis, and anemia).

**Sample size and sampling technique.** The sample size was calculated using a single population proportion formula of hypogonadism with Open-Epi software. The values of the outcome variable were taken from the previous study conducted in South Africa [32]. Taking the population proportion to be 73% and assuming a 95% confidence interval, after doubling the sample size, the minimal sample size in each group is expected to be 303. The final sample size was corrected with the reduction formula due to the previous two months' average TB cases being 261 in UGRCSH and health centers. From this, 70 were in UGRCSH, 58 in Marakie, 52 in Azezo, 41 in Poli, and 40 in the Mitwab health center.

$$\text{sample size} = n_0/((1+(n_0)/N) = 303/((1+(303)/261) = 140 = 140*2 = 280$$

After adding a 10% non-response rate, the final sample size was 308 (154 confirmed TB-positive patients and 154 apparently healthy TB-negative comparison group). A totals 300 study participants with response rate 97.4% were recruited on the present study.

After proportionally allocating the cases, different numbers of study participants were recruited from the study settings (41, 34, 31, 24, and 24 from UGRCSH, Marakie, Azezo, Poli, and Mitwab health centers, respectively). A simple random (table of random number) sampling technique was used to recruit the TB-positive patients. A convenient sampling technique was used to recruit the TB-negative comparison group and for selecting study hospitals and health centers.

## Data collection

The sociodemographic, anthropometric, behavioral, and clinical data were collected using a pre-tested structured questionnaire that was translated from English to Amharic and administered through face-to-face interviews. Trained clinical nurses screened participants for eligibility and obtained written informed consent.

The weight of the participant was measured using a calibrated Seca digital weighing scale (Seca, Hamburg, Germany) while standing upright with feet evenly spaced apart, distributing weight evenly on both feet to ensure balance. Measurements were taken with participants barefoot and wearing light clothing, and weight was recorded to the nearest 0.1 kg. Using a standard and calibrated stadiometer (Seca, Hamburg, Germany), the height of participants was measured by ensuring that their body was properly aligned, with the head level, eyes looking straight ahead, shoulders relaxed, and arms hanging naturally by their sides. Height was recorded to the nearest 0.5 cm. BMI was calculated as weight in kilograms divided by height in meters squared and categorized as underweight (<18.5 kg/m²), normal (18.5–24.9), overweight (25.0–29.9), and obese (≥ 30) [33,34].

Blood pressure was measured on the left arm using a mercury sphygmomanometer (India) with a standardized automatic blood pressure monitor. Participants remained seated and relaxed for at least five minutes before measurements. Blood pressure was determined by averaging two measurements of systolic blood pressure (SBP) and diastolic blood pressure (DBP). If the difference between the two readings exceeded 10 mmHg, a third reading was taken, and the average of the three measurements was recorded. Blood pressure was categorized as normal (SBP<120 mmHg and DBP<80 mmHg), prehypertension (SBP 120–139 mmHg or DBP 80–89 mmHg), and hypertension (SBP≥140 mmHg and/or DBP≥90 mmHg) [35].

Behavioral data (smoking, alcohol consumption, physical activity, and dietary diversity) were collected through interviews. Smoking status was categorized as non-smoker (never smoked), current smoker (smoked within the last 30 days),

or former smoker (quit prior to the interview) [36,37]. Alcohol consumption was categorized as non-consumer (never consumed alcohol), current consumer (equivalent to 2–4 standard drinks per week in the past 30 days), or former consumer (stopped drinking prior to the interview) [37]. Physical activity was defined as engaging in ≥150 minutes per week of moderate-intensity activity or ≥3 days per week of vigorous-intensity activity [34]. High dietary diversity was defined as the regular intake of staple grains, fruits, vegetables, meat, fish, eggs, milk, and iodized salt [38]. Clinical data were extracted from medical charts, including TB classification (pulmonary vs. extrapulmonary), treatment status, treatment duration (classified into two groups: those on treatment for 2 months or less and those on treatment for more than 2 months), and comorbidities (S1 Table).

## Laboratory procedures and analysis

Following face-to-face interviews with structured questions, fasting venous blood samples were collected between 8:00 AM and 10:00 AM from each participant after a minimum of an eight-hour fast and processed according to standard procedures of the hospital's laboratory. Five milliliters of blood were drawn from the medial cubital vein of the left arm using a 19-gauge syringe and then transferred to gel-coated serum separator tubes labeled with unique ID numbers.

The blood was allowed to clot at room temperature for 30 minutes, then centrifuged at 3000 RPM for 5 minutes to obtain serum. The sex hormone profile and cortisol were analyzed from serum using a chemiluminescent immunoassay on the Beckman Coulter DXI 800 automated clinical chemistry analyzer (DXI800, Beckman Coulter Inc, Danaher Corporation Company Brea, California, United States). After running the test, the result was recorded on data collection sheets and subsequently entered into statistical software.

## Data quality control

Prior to data collection, data collectors underwent comprehensive training on study objectives, ethical considerations, consent procedures, interview techniques, and laboratory protocols. The questionnaire was pretested on 5% of the sample size to ensure accuracy and consistency. Laboratory analysis was carried out using a Beckman-Coulter DXI 800 chemistry analyzer, with commercial quality control materials run alongside patient specimens to ensure precision and accuracy. The primary investigator closely supervised all data collection and laboratory procedures to maintain data integrity. Collected data were regularly checked for errors, corrected on the same day, if necessary, securely recorded, and properly stored until entry into statistical software. This manuscript was adhered to STROBE checklist and provide a completed STROBE checklist (S2 Data).

## Statistical analysis

All participant data was first entered into EPI data version 4.6.0.2 for data clearance and consistency, followed by transfer to SPSS version 25.0 (IBM Corp., Armonk, NY, USA) for statistical analysis. Descriptive statistics were computed to characterize the data, while the Kolmogorov-Smirnov test was used to check the normal distribution of continuous variables. Continuous variables that are normally distributed were summarized using the mean and standard deviations, whereas variables that were non-normally distributed were summarized using their median and interquartile ranges. The independent t-test, Mann-Whitney U, one-way ANOVA with Tukey HSD post-hoc, and Kruskal-Wallis H test with Dunn's post-hoc were used for comparison of sex hormone profiles within the groups, and a logistic regression model was fitted to identify associated factors, with statistical significance considered at $p < 0.05$ in the multivariable model and $p < 0.25$ in the bivariable model. Prior to logistic regression analysis, a rigorous variable screening process was implemented to select categorical independent variables. The independence and exclusivity of each variable were confirmed, and variables exhibiting multicollinearity were excluded from the analysis. Categorical independent variables were assessed using the chi-square $(X^2)$, with those meeting assumptions subjected to bivariate logistic regression analysis. The logistic regression model was fitted, and the Hosmer-Lemeshow test was analyzed to test the goodness of fit of the logistic regression model. Finally, a P value $< 0.05$ was considered as statistically significant.

## Operational definitions

**Male HG.** Primary HG: low testosterone (< 3.461 ng/ml) with high LH and FSH.

Secondary HG: low testosterone, LH, and FSH.

Subclinical HG: low testosterone with normal LH and FSH [39,40].

**Female HG.** Defined using reference ranges from the Beckman Coulter analyzer at UGRCSH [41]. Primary HG: low estradiol and progesterone, high FSH and LH.

Secondary HG: low estradiol and progesterone, low FSH and LH.

Subclinical HG: low estradiol and progesterone, normal FSH and LH.

**Menstrual cycle.** regular cycle every 21–35 days lasting 2–7 days; variability in these metrics was considered as a menstrual irregularity [42].

**Tuberculosis -classification.** PTB if involving the lungs; EPTB if involving other organs [43].

**Tuberculosis-negative comparison group.** Apparently healthy, TB-negative individuals without signs or symptoms of TB.

## Ethical consideration

The study was conducted in accordance with the Declaration of Helsinki guidelines and regulations. It was carried out after ethical approval was obtained from the Institutional Review Board of the School of Biomedical and Laboratory Sciences, College of Medicine and Health Sciences, University of Gondar, with reference number SBMLS/763, before commencing data collection. Permission was also gained from the medical director of UGRCSH and each of the health centers. Informed consent was obtained from each participant after appropriate information was given about the aim of the project and objective of the study, and confidentiality of data was maintained throughout the study. The collected data was not used for another purpose other than the present study.

## Results

### Sociodemographic characteristics of study participants

A total of 150 TB-positive patients and 150 TB-negative participants as a comparison group matched by age were included in the present study. Of these 162 (54%) were males (81 TB-positive patients and 81 TB-negative comparison group), aged between 18 and 76 years, and 138 (46%) were females (69 TB positive-patients and 69 TB-negative comparison group), aged between 18 and 53 years. The mean age of TB-positive patients was 32.86 ± 12.88, and 32.81 ± 12.2 was for the TB-negative comparison group (Table 1).

### Clinical and behavioral characteristics of the study participants

The mean BMI of TB-positive patients was 19.29 ± 2.75, while the mean BMI for the TB-negative comparison group was 21.14 ± 2.59. Among 115 female study participants (55 TB-positive patients and 60 TB-negative comparison group), the overall prevalence of menstrual irregularities among participants was 26 (22.6%). The prevalence of menstrual irregularities was higher among TB-positive patients, 22 (40%), compared to the TB-negative comparison group, 4 (6.7%). In addition, out of 138 female participants, 23 reported cessations of menstruation (Table 2).

### Comparison of sex hormone profiles among study participants

In the current study, involving 81 male TB-positive patients and 81 male TB-negative comparison groups, TB-positive patients had significantly higher mean values for estradiol (p < 0.001), LH (p < 0.001), and FSH (p < 0.001); however, they had significantly lower testosterone (p < 0.001) compared to the comparison group, as shown by the independent t-test. In the female participants analysis, through the Mann-Whitney U Test with a total of 138 cases, comprising 69 TB-positive

**Table 1. Sociodemographic characteristics among study participants (n = 300).**

| Variables | Categories | TB-positive patients | TB negative group | Total | P value |
|---|---|---|---|---|---|
| | | n (%) | n (%) | n (%) | |
| **Sex** | Male | 81 (54.0%) | 81 (54.0%) | 162 (54.0%) | 1 |
| | Female | 69 (46.0%) | 69 (46.0%) | 138 (46.0%) | |
| **Age (years)** | 18-25 | 49 (32.7%) | 50 (33.3%) | 99 (33.0%) | 0.999 |
| | 26-35 | 53 (35.3%) | 52 (34.7%) | 105 (35.0%) | |
| | 36-45 | 23 (15.3%) | 23 (15.3%) | 46 (15.3%) | |
| | >45 | 25 (16.7%) | 25 (16.7%) | 50 (16.7%) | |
| **Marital status** | Single | 52 (34.0%)? | 85 (56.7%) | 136 (45.3%) | <0.001* |
| | Married | 82 (54.7%) | 43 (28.7%) | 125 (41.7%) | |
| | Widowed | 9 (6.0%) | 10 (6.7%) | 19 (6.3%) | |
| | divorced | 8 (5.3%) | 12 (8.0%) | 20 (6.7%) | |
| **Monthly Income (ETB)** | ≤3000 (ETB) | 86 (57.3%) | 63 (42.0%) | 149 (49.7%) | 0.008* |
| | > 3000 (ETB) | 64 (42.7%) | 87 (58.0%) | 151 (50.3%) | |
| **Education** | Unable to read and write | 36 (24.0%) | 4 (2.7%) | 40 (13.3%) | <0.001* |
| | Primary school | 42 (28.0%) | 24 (16.0%) | 66 (22.0%) | |
| | Secondary school | 45 (30.0%) | 30 (20.0%) | 75 (25.0%) | |
| | Higher education | 27 (18.0%) | 82 (61.3%) | 119 (39.7%) | |
| **Residency** | Urban | 119 (86.0%) | 142 (94.7%) | 271 (90.3%) | 0.011* |
| | Rural | 21 (14.0%) | 8 (5.3%) | 29 (9.7%) | |
| **Occupation** | House wife | 33 (22.0%) | 15 (10.0%) | 48 (16.0%) | <0.001* |
| | Employed | 59 (39.3%) | 83 (55.3%) | 142 (47.3%) | |
| | Student | 23 (15.3%) | 34 (22.7%) | 57 (19.0%) | |
| | Unemployed | 21(14.0%) | 18 (12.0%) | 39 (13.0%) | |
| | Laborer | 14 (9.4%) | – | 14 (4.7%) | |

* = statistically significant at P <0.05, ETB = Ethiopian Birr.

patients and 69 TB-negative comparison group. TB-positive patients had significantly lower median serum testosterone (p < 0.001) and progesterone (p < 0.001) levels compared to the TB-negative comparison group (Table 3).

### Sex hormone profiles in newly diagnosed TB-positive patients, TB-positive patients on treatment, and a TB-negative comparison group

Among Males: Statistically significant differences were observed in all sex hormones (testosterone, estradiol, progesterone, LH, and FSH) across the three groups in the ANOVA table below (p < 0.05);. Post hoc Tukey HSD analysis showed that newly diagnosed TB-positive patients had significantly lower mean testosterone levels compared with TB-positive patients on treatment (p = 0.002) and the TB-negative comparison group (p < 0.001). Similarly, progesterone levels were significantly lower in newly diagnosed TB-positive patients compared with TB-positive patients on treatment (p = 0.004) and the TB-negative comparison group (p = 0.02). In contrast, estradiol levels were significantly higher in newly diagnosed TB-positive patients compared with both TB-positive patients on treatment (p < 0.001) and the TB-negative comparison group (p < 0.001).

Among Females: The Kruskal–Wallis H test revealed statistically significant differences in median testosterone, estradiol, progesterone, and FSH across the three groups (p < 0.001);. Dunn's post hoc analysis showed that both newly diagnosed TB-positive patients and TB-positive patients on treatment had significantly lower median testosterone levels compared with the TB-negative comparison group (p < 0.001). In contrast, newly diagnosed TB-positive patients had significantly higher median

**Table 2. Clinical and behavioral characteristics among study participants (n = 300).**

| Variable | Categories | | TB-positive patients | TB-negative group | Total | P value |
|---|---|---|---|---|---|---|
| | | | n (%) | n (%) | n (%) | |
| BMI (kg/m²) | Under weight | | 66 (44.0%) | 26 (17.3%) | 92 (30.7%) | <0.001* |
| | Normal weight | | 80 (53.3%) | 112 (74.7%) | 192 (64.0%) | |
| | Over weight | | 4 (2.7%) | 12 (8.0%) | 16 (5.3%) | |
| Blood pressure(mmHg) | SBP | ≤120 mmHg | 140 (93.3%) | 141 (94.0%) | 281 (93.7%) | 0.813 |
| | | >120mmHg | 10 (6.7%) | 9 (6.0%) | 19 (6.3%) | |
| | DBP | ≤80 mmHg | 133 (88.7%) | 132 (88.0%) | 265 (88.3%) | 0.857 |
| | | >80mmHg | 17 (11.3%) | 18 (12.0%) | 35 (11.7%) | |
| Cortisol (ng/ml) | Below the mean | | 53 (35.3%) | 120 (80.0%) | 173 (57.7%) | <0.001* |
| | Above the mean | | 97 (64.7%%) | 30 (20.0%) | 127 (42.3% | |
| Family history of medical disease | Yes | | 19 (12.7%) | 14 (9.3%) | 33 (11.0%) | 0.356 |
| | No | | 131 (87.3%) | 136 (90.7%) | 267 (89.0%) | |
| Underline medical condition | Yes | | 7 (4.7%) | – | 7 (2.3%) | NA |
| | No | | 143 (95.3%) | 150 (100.0%) | 293 (97.7%) | |
| Alcohol drinking | No | | 130 (86.7%) | 137 (91.3%) | 267 (89.0%) | 0.001* |
| | Yes | | 2 (1.3%) | 9 (6.0%) | 11 (3.7%) | |
| | Stopped | | 18 (12.0%) | 4 (2.7%) | 22 (7.3%) | |
| Smoking habit | No | | 148 (98.7%) | 150 (100.0%) | 298 (99.3%) | NA |
| | Yes | | – | – | – | |
| | stopped | | 2 (1.3%) | – | 2 (0.7%) | |
| Physical activity | Yes | | 21 (14.0%) | 16 (10.7%) | 37 (12.3%) | 0.38 |
| | No | | 129 (86.0%) | 134 (89.3%) | 263 (87.7%) | |
| Dietary diversity (high diet diversity) | No | | 24 (16.0%) | 5 (3.3%) | 29(10.0%) | <0.001* |
| | Sometimes | | 41 (27.3%) | 27 (18.0%) | 68 (22.7%) | |
| | Usually | | 39 (26.0%) | 84 (56.0%) | 123 (41.0%) | |
| | Always | | 46 30.7%) | 34 (27.7%) | 80 (26.7% | |
| Usual food source | Plant/ Vegetable | | 66 (44.0%) | 37 (24.7%) | 103 (34.3%) | <0.001* |
| | Animal and dairy products | | 37 (24.7%) | 22 (14.7%) | 59 (19.7%) | |
| | Both | | 47 (31.3%) | 91 (60.7%) | 138 (46.0%) | |
| Menstrual Irregularity | Irregular | | 22 (40.0%) | 4 (6.7%) | 26 (22.6%) | 0.077 |
| | Regular | | 33 (60.0%) | 56 (93.3%) | 89 (77.4%) | |

Clinical Profile and Tuberculosis related sign and symptoms

| Variable | Categories | Male (81) | Female (69) | total | P value |
|---|---|---|---|---|---|
| | | n (%) | n (%) | | |
| Vomit | Yes | 22 (27.2%) | 35 (50.7%) | 57 (38.0%) | 0.003* |
| | No | 59 (72.8%) | 34 (49.3%) | 93 (62.0%) | |
| Nausea | Yes | 33 (40.7%) | 35 (50.7%) | 68 (45.3%) | 0.221 |
| | No | 48 (59.3%) | 34 (49.3%) | 82 (54.7%) | |
| Loss of appetite | Yes | 34 (42.0%) | 22 (31.9%) | 56 (37.3%) | 0.203 |
| | No | 47 (58.0%) | 47 (68.1%) | 94 (62.7%) | |
| Previous history of TB | Yes | 6 (7.4%) | 7 (10.1%) | 13 (8.7%) | 0.553 |
| | No | 75 (92.6%) | 62 (89.9%) | 137 (91.3%) | |

*(Continued)*

**Table 2.** (Continued)

| Variable | Categories | Male (81) | Female (69) | total | P value |
|---|---|---|---|---|---|
| | | n (%) | n (%) | | |
| **TB class** | PTB | 58 (71.6%) | 38 (55.1%) | 96 (64.0%) | 0.036* |
| | EPTB | 23 (28.4%) | 31 (44.9%) | 54 (36.0%) | |
| **Treatment status** | New | 29 (35.8%) | 24 (34.8%) | 53 (35.3%) | 0.896 |
| | On treatment | 52 (64.2%) | 45 (65.2%) | 97 (64.7%) | |
| **Medication other than anti-TB drugs** | Yes | 5 (6.2%) | 2 (2.9%) | 7 (4.7%) | NA |
| | No | 76 (93.8%) | 67 (97.1%) | 143 (95.3%) | |

Note: *=statistically significant at P<0.05, NA – not applicable.

**Table 3.** Comparison of sex hormone profile among study participants (162 male and 138 female).

| Parameters | | Testosterone (ng/ml) | | Estradiol (pg/ml) | P4 (ng/ml) | LH (mIU/ml) | FSH (mIU/ml) |
|---|---|---|---|---|---|---|---|
| **Male** | TB-positive patients | X±SD | 3.3±1.54 | 48.2±23.94 | 0.4±0.3 | 11.7±5.5 | 12.8±5.6 |
| | TB-negative group | X±SD | 5.9±1.4 | 33.1±8.5 | 0.4±0.2 | 5.9±4.7 | 6.2±3.5 |
| | (95% CI) | (−3.1, −2.1) | | (9.5, 20.7) | (−0.1, 0.1) | (3.2, 8.3) | (4.1, 9.0) |
| | Mean difference | −2.6 | | +15.1 | −0.1 | +5.8 | +6.6 |
| | P value | <0.001* | | <0.001* | 0.531 | <0.001* | <0.001* |
| **Female** | TB-positive patients | M±IQR | 0.2±0.1 | 80.4±200.6 | 0.4±1.4 | 10.8±14.6 | 8.4±16.6 |
| | TB-negative group | M±IQR | 0.6±0.2 | 69.6±56.2 | 1.5±11.5 | 8.9±15.3 | 5.7±4.5 |
| | Effect size (r) | 0.7 | | 0.1 | 0.4 | 0.2 | 0.3 |
| | P value | <0.001* | | 0.28 | <0.001* | 0.079 | <0.001* |

**Abbreviations:** X±SD (mean plus or minus standard deviations), M±IQR (median plus or minus interquartile range) CI (Confidence Interval) Note: *=statistically significant at P<0.05.

"+" Increase among TB-positive patients "-" Decreased among TB-positive patients.

estradiol levels compared with both TB-positive patients on treatment and the TB-negative comparison group (p<0.001). Moreover, progesterone levels were significantly lower in TB-positive patients on treatment compared with the TB-negative comparison group (p<0.001). Finally, the TB-negative comparison group had significantly lower median FSH levels compared with newly diagnosed TB-positive patients (p=0.04) and TB-positive patients on treatment (p<0.001) (Table 4).

## Comparison of sex hormones according to treatment duration among TB-positive patients

Male sex hormones: testosterone, estradiol, and progesterone levels had shown statistically significant mean differences (p<0.05) across groups as described in the ANOVA table below. The post hoc Tukey HSD analysis described as the following:

Newly diagnosed TB-positive patients had significantly lower mean testosterone levels compared to both TB-positive patients on treatment for 2 months or less (p=0.02) and more than 2 months (p=0.003). However, the mean serum estradiol levels were significantly higher than those of TB-positive patients on treatment for 2 months or less (p<0.001) and more than 2 months (p=0.001). Finally, newly diagnosed TB-positive patients had significantly lower mean progesterone levels than those treated for 2 months or less (p=0.01).

In females, serum testosterone, estradiol, and progesterone levels had statistically significant median differences across groups as described in the Kruskal-Wallis H test below. The post hoc Dunn's analysis yielded the following:

Those treated for more than 2 months had significantly higher median testosterone levels than newly diagnosed TB-positive patients (p<0.001) and those treated for 2 months or less (p=0.001). However, newly diagnosed TB-positive

**Table 4. Sex Hormone Profiles among newly diagnosed TB-positive patients, TB-positive patients on treatment, and TB-negative comparison group).**

| Parameters | | | Testosterone (ng/ml) | Estradiol (pg/ml) | P4 (ng/ml) | LH (mIU/ml) | FSH (mIU/ml) |
|---|---|---|---|---|---|---|---|
| Male (162) | New TB patients | X±SD | (2.5±1.3) [a] | (62.8±25.8) [a] | (0.3±0.2) [a] | (8.8±3.3) [a] | 13.3±5.7 |
| | On Rx | X±SD | (3.7±1.53) [b] | (40.0±18.6) [b] | (0.5±0.3) | (13.3±12.7) [b] | (12.5±12.5) [b] |
| | TB-negative group | X±SD | (5.9±1.4) [c] | (33.1±8.5) [c] | (0.4±0.2) [c] | (5.9±4.7) | (6.2±3.5) [c] |
| | P value | | <0.001* | <.001* | 0.005* | <0.001* | <0.001* |
| Female (138) | New TB patients | M±IQR | (0.2±0.1) | (221.2±228.0) | (1.1±10.8) [a] | 12.3±13.6 | 8.3±27.5) |
| | On RX | M±IQR | (0.2±0.2) [b] | (43.8±113.7) [a] | (0.3±0.6) [b] | 9.7±13.2 | (8.4±16.4) [b] |
| | TB-negative group | M±IQR | (0.6±0.2) [c] | (69.6±56.2) [c] | 1.5±11.5 | 9.0±15.3 | (5.7±4.5) [c] |
| | P value | | <0.001* | <0.001* | <0.001* | 0.098 | <0.001* |

Note: [a] = comparison between new cases and cases on Rx, [b] = comparison between cases on Rx and controls,

[c] = comparison between new cases and controls, * = statistically significant at P<0.05.

New = newly diagnosed TB-positive patients, RX = TB-positive patients on treatment.

patients had significantly higher median estradiol than those treated for 2 months or less (p = 0.012) and more than 2 months (p = 0.005) (Table 5).

## Prevalence and type of hypogonadism among study participants

Among study participants, the overall prevalence of HG was 91 (30.3%) (95% CI 25.2–35.9%). The prevalence of primary, secondary, and subclinical HG was 37 (12.3%, 95% CI 8.8–16.6%), 12 (4%, 95% CI 2.1–6.9%), and 42 (14%, 95% CI 10.3–18.4%), respectively (Fig 1).

The prevalence of HG was significantly higher among TB-positive patients compared to the TB-negative comparison group (*p*<0.001). In TB-positive patients the prevalence of HG was 73 (48.7%, 95% CI 40.4%−57.0%), while in the TB-negative comparison group it was 18 (12%, 95% CI 7.3–18.3%) (Fig 2).

In the current study, the prevalence of HG increased from 25.3% (25 out of 99) among 18–25 years old study participants to 50% (25 out of 50) among participants aged over 45 years old (Fig 3).

**Table 5. Sex hormones according to treatment duration among TB-positive patients.**

| Parameters | | | Testosterone (ng/ml) | Estradiol (pg/ml) | P4 (ng/ml) | LH (mIU/ml) | FSH (mIU/ml) |
|---|---|---|---|---|---|---|---|
| **Male (81)** | New TB patients | X±SD | (2.5±1.3) [a] | (62.8±25.8) [a] | (0.3±0.2) [a] | 8.8±3.3 | 13.3±5.7 |
| | On Rx ≤2 month | X±SD | 3.5±1.6 | 40.7±19.3 | 0.5±0.4 | 13.2±11.6 | 11.7±12.6 |
| | On Rx >2 month | X±SD | (4.0±1.4) [c] | (38.7±17.6) [c] | 0.4±0.2 | 13.7±14.7 | 14.0±12.6 |
| | P value | | 0.002* | <0.001* | 0.013* | 0.171 | 0.717 |
| **Female (69)** | New TB patients | M±IQR | (0.15±0.1) [c] | (221.2±228.0) | (1.1±10.8) [a] | 12.3±13.6 | 8.3±27.5 |
| | On RX ≤2 month | M±IQR | (0.2±0.1) [b] | 59.0±131.3 | 0.3±0.7 | 10.1±20.2 | 8.6±16.5 |
| | On Rx >2 month | M±IQR | 0.4±0.3 | 27±59.0 | 0.3±0.6 | 9.5±6.7 | 8.2±22.8 |
| | P value | | <0.001* | 0.002* | 0.014* | 0.308 | 0.936 |

Note: [a] = comparison between new cases and cases on Rx for 2 months or less, [b] = comparison between cases on Rx for 2 months or less and On Rx for more than 2 months,

[c] = comparison between new cases and On Rx for more than 2 months, * = statistically significant at P<0.05.

New = newly diagnosed TB-positive patients, RX = TB-positive patients on treatment.

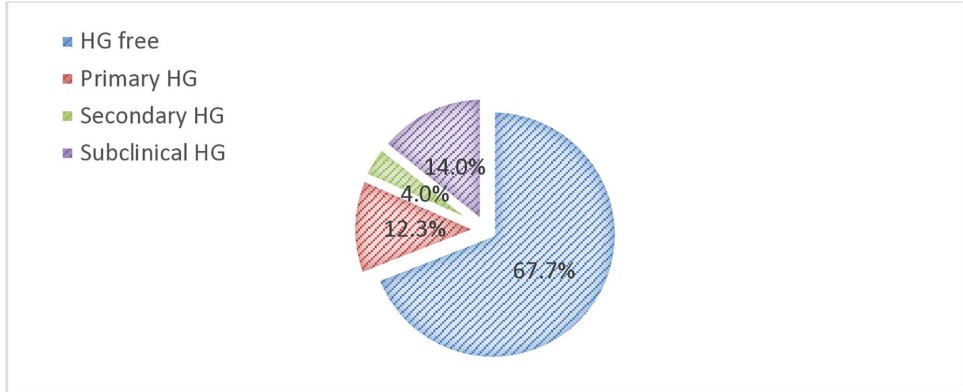

**Fig 1. Prevalence of HG among overall study participants attending.**

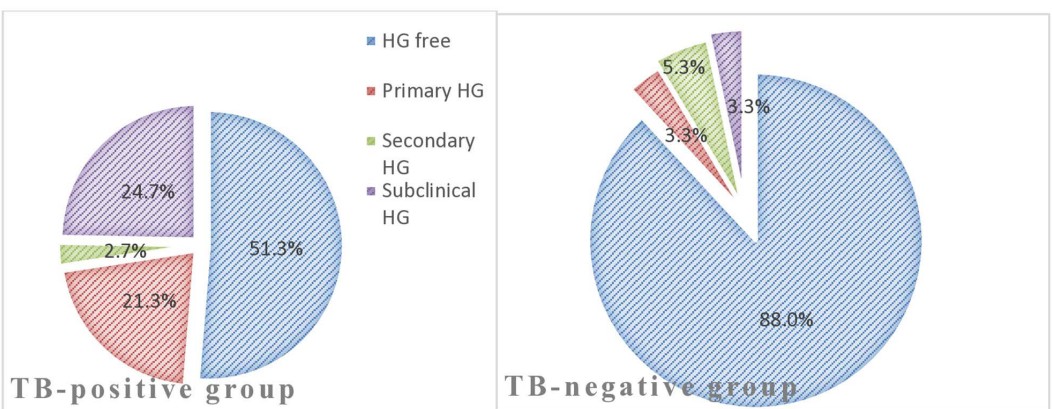

**Fig 2. Prevalence of HG among TB-positive patients and the TB-negative comparison group.**

## Factors associated with hypogonadism among study participants

Independent variables with a p-value<0.25 in the bivariable logistic regression model were taken into the multivariable logistic regression model. Accordingly, TB status, alcohol consumption, and cortisol levels were the determinant factors of HG among study participants on the multivariable logistic regression model.

Tuberculosis-positive patients had 8.37 times higher odds of developing HG when compared to the TB-negative comparison group [AOR=8.37, 95% CI (2.99, 23.4)], ($p<0.001$). Participants with a history of alcohol consumption had 4.1 times higher odds of having HG when compared to participants who did not drink [AOR=4.1, 95% CI (1.21, 13.95)], ($p=0.024$). Individuals with high values of cortisol (above the mean value) had a significantly increased risk of developing HG, with odds being 3.4 times higher compared to those with cortisol levels below the mean value [AOR=3.4, 95% CI (1.55, 7.46)], ($p=0.002$) (Table 6).

## Factors associated with hypogonadism among TB-positive patients

Independent variables with a p-value<0.25 in the bivariable logistic regression model were taken into the multivariable logistic regression model. This revealed that sex, dietary diversity, and cortisol levels were the significant determinant factors of HG among TB-positive patients on the multivariable logistic regression model.

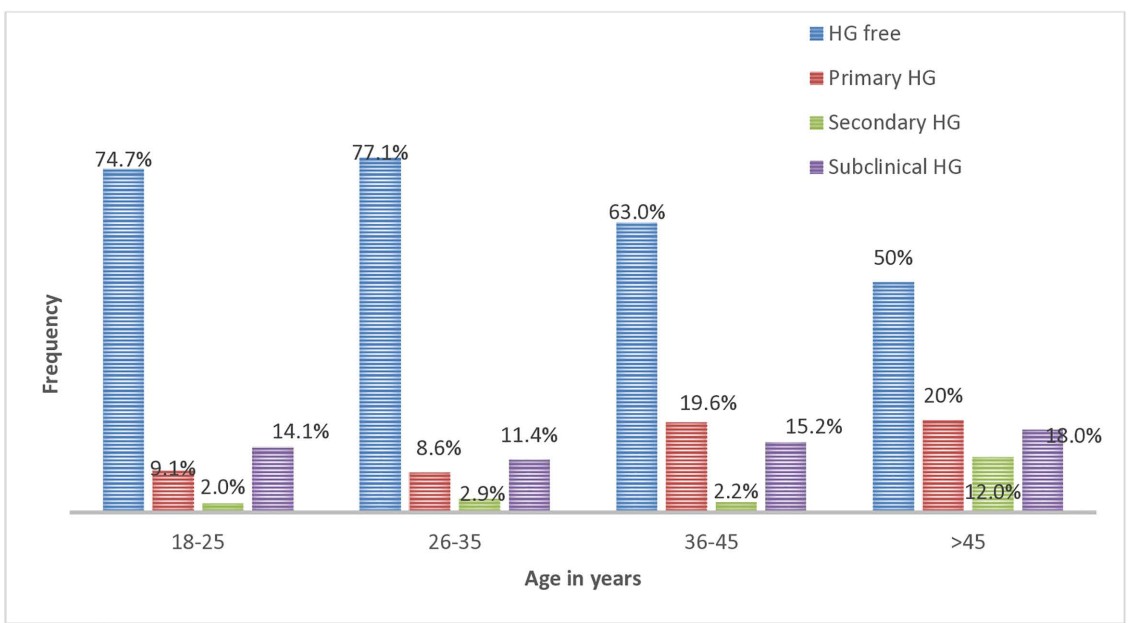

**Fig 3. Prevalence of HG with age among study participants.**

This study found that male TB-positive patients had 11.36 times higher odds of HG compared to female participants [AOR = 11.36, 95% CI (3.6, 36.17)], ($p < 0.001$). Additionally, dietary diversity was the other factor, with participants consuming diets with lower diversity, including those with no dietary diversity [AOR = 8.98, 95% CI (2.37, 33.99)] ($p = 0.001$), those with sometimes [AOR = 9.2, 95% CI (2.77, 30.62)] ($p < 0.001$), and those with a usual dietary diversity [AOR = 3.24, (1.04, 10.06) ($p = 0.042$) also having 8.98, 9.2, and 3.24 times greater odds of HG compared to those with always diverse diets respectively. Furthermore, Participants with serum cortisol levels above the mean value had 4.01 times greater odds of HG compared to those with serum cortisol below the mean value [AOR = 4.01, 95% CI (1.7, 9.5)], ($p = 0.002$) (Table 7).

## Discussion

Tuberculosis disproportionately affects adults, with 90% of infections occurring in individuals aged 18 years and older, highlighting its significant global health burden [44]. To the best of our searching effort, this was the first study to investigate the serum sex hormone levels in adult patients with TB in Gondar.

In the present study, male TB-positive patients had significantly lower testosterone levels compared to the TB-negative comparison group; this finding aligns with previous studies conducted in Egypt [45], India [46], and Mexico [19]. TB-positive patients showed lower testosterone despite higher LH and FSH levels. FSH and LH cooperate to cause the testes to produce sperm and testosterone. LH from the anterior pituitary acts on Leydig cells, binds with receptors on the surface of Leydig cells, and triggers the synthesis of testosterone [47]. The uncoupling of LH and testosterone suggests either resistance or an inhibition of Leydig cell steroidogenesis mediated by cytokines that impinge on this process [19]. Infectious stimuli may be connected to these phenomena. Because TB patients' testes had higher concentrations of pro-inflammatory cytokines, these cytokines dramatically reduced Leydig cells' ability to produce testosterone [17]. However, this finding is contrary to the report of a previous study in Ethiopia [48], which reported no significant difference in plasma testosterone level between groups. This difference may be due to the difference in method used for measuring testosterone, and the sample for analysis was conducted on stored samples from the Armauer Hansen Research Institute

**Table 6. Bivariable and multivariable logistic regression analysis of factors associated with HG among study participants (n=300).**

| Variable | Categories | Hypogonadism | | COR (95%CI) | AOR (95% CI) | P value |
|---|---|---|---|---|---|---|
| | | Yes n (%) | No n (%) | | | |
| **TB status** | TB-positive patients | 73 (48.7%) | 77 (51.3%) | 7.0 (3.9, 12.5) | 8.4 (3.0, 23.4) | <0.001* |
| | TB-negative comparison group | 18 (12.0%) | 132 (88.0%) | 1 | | |
| **Sex** | Male | 57 (35.2%) | 105 (64.8%) | 1.7 (1.0, 2.8) | 1.9 (0.7, 4.6) | 0.187 |
| | Female | 34 (24.6%) | 104 (75.4%) | 1 | | |
| **Age (years)** | 18-25 (years) | 25 (25.3%) | 74 (74.7%) | 1 | | |
| | 26-35 (years) | 24 (22.9%) | 81 (77.1%) | 0.9 (0.5, 1.7) | 0.5 (0.2, 1.6) | 0.227 |
| | 36-45 (years) | 17 (37.0%) | 29 (63.0%) | 1.7 (0.8, 3.7) | 1.4 (0.4, 5.3) | 0.595 |
| | >45 (years) | 25 (50.0%) | 25 (50.0%) | 3.0 (1.5, 6.1) | 1.7 (0.4, 6.7) | 0.481 |
| **Marital status** | Single | 27 (19.9%) | 109 (80.1%) | 1 | | |
| | Married | 46 (36.8%) | 79 (63.2%) | 2.4 (1.4, 4.1) | 1.07 (0.4, 2.7) | 0.889 |
| | Widowed | 11 (57.9%) | 8 (42.1%) | 5.6 (2.0, 15.1) | 5.1 (1.0, 26.5) | 0.051 |
| | Divorced | 7 (35.0%) | 13 (65.0%) | 2.2 (0.8, 6.0) | 3.0 (0.7, 13.2) | 0.145 |
| **Educational status** | Unable to read and write | 20 (50.0%) | 20 (50.0%) | 5 (2.3, 10.8) | 1.9 (0.4, 9.1) | 0.419 |
| | Primary school | 27 (40.9%) | 39 (59.1%) | 3.4 (1.7, 6.8) | 2.4 (0.8, 7.0) | 0.121 |
| | Secondary school | 24 (32.0%) | 51 (68.0%) | 2.3 (1.2, 4.6) | 1.5 (0.5, 3.9) | 0.462 |
| | Higher education | 20 (16.8%) | 99 (83.2%) | 1 | | |
| **Residency** | Rural | 13 (44.8%) | 16 (55.2%) | 2.0 (0.9, 4.4) | 2.9 (0.9, 9.4) | 0.078 |
| | Urban | 78 (28.8%) | 193 (71.2%) | 1 | | |
| **Occupation** | House wife | 17 (35.4%) | 31 (64.6%) | 1 | | |
| | Employed | 47 (33.1%) | 95 (66.9%) | 0.9 (0.5, 1.8) | 1.4 (0.4, 5.0) | 0.654 |
| | Student | 11 (19.3%) | 46 (80.7%) | 0.4 (0.18, 1.1) | 1.5 (0.3, 6.7) | 0.635 |
| | Unemployed | 11 (28.2%) | 28 (71.8%) | 0.8 (0.3, 1.8) | 1.3 (0.1, 5.8) | 0.724 |
| | Laborer | 5 (35.7%) | 9 (64.3%) | 1.0 (0.3, 3.5) | 0.4 (0.1, 2.0) | 0.244 |
| **Monthly Income (ETB)** | ≤3000 (ETB) | 38 (25.5%) | 111 (74.5%) | 0.6 (0.4, 1.0) | 0.4 (0.2, 1.1) | 0.08 |
| | >3000 (ETB) | 53 (35.1%) | 98 (64.9%) | 1 | | |
| **BMI (kg/m²)** | Normal weight | 54 (28.1%) | 138 (71.9%) | 1 | | |
| | Underweight | 35 (38.0%) | 57 (62.0%) | 1.6 (0.9, 2.7) | 0.8 (0.4, 1.7) | 0.519 |
| | Overweight | 2 (12.5%) | 14 (87.5%) | 0.4 (0.1, 1.7) | 0.7 (0.1, 3.7) | 0.631 |
| **SBP (mmHg)** | ≤120 (mmHg) | 82 (29.2%) | 199 (70.8%) | 1 | | |
| | >120 (mmHg) | 9 (47.4%) | 10 (52.6%) | 2.2 (0.9, 5.6) | 1.1 (0.3, 4.1) | 0.861 |
| **Alcohol drinking** | No | 73 (27.3%) | 194 (72.7%) | 1 | | |
| | Yes | 4 (36.4%) | 7 (63.6%) | 1.5 (0.4, 5.3) | 1.8 (0.4, 9.2) | 0.464 |
| | stopped | 14 (63.6%) | 8 (34.4%) | 4.7 (1.9, 11.6) | 4. (1.2,14.0) | 0.024* |
| **Dietary diversity (high diet diversity)** | No | 16 (55.2%) | 13 (44.8%) | 3.7 (1.5, 9.0) | 2.7 (0.8, 9.2) | 0.103 |
| | Sometimes | 18 (26.5%) | 50 (73.5%) | 1.1 (0.5, 2.3) | 1.5 (0.5, 4.5) | 0.445 |
| | Usually | 37 (30.1%) | 86 (69.9%) | 1.3 (0.6, 2.4) | 4.8 (0.9,12.1) | 0.052 |
| | Always | 20 (25.0%) | 60 (75.0%) | 1 | | |
| **Usual food source** | Plant/vegetable | 36 (35.0%) | 67 (65.0%) | 1.7 (1.0, 3.0) | 1.1 (0.4, 2.5) | 0.905 |
| | Animal and dairy products | 22 (37.3%) | 37 (62.7%) | 1.9 (1.0, 3.7) | 1.1 (0.5, 2.8) | 0.792 |
| | Both | 33 (23.9%) | 105 (76.1%) | 1 | | |
| **Cortisol levels (ng/ml)** | Below the mean | 32 (18.5%) | 142 (81.5%) | 1 | | |
| | Above the mean | 59 (46.5%) | 68 (53.5%) | 3.8 (2.3, 6.4) | 3.4 (1.6, 7.5) | 0.002* |

**Abbreviations:** AOR (Adjusted Odds Ratio), COR (Crude Odds Ratio), CI (Confidence Interval)

**Note:** * = Statistically Significant at P <0.05, 1 = Reference Group, ETB = Ethiopian Birr

**Table 7. Bivariable and multivariable logistic regression analysis of factors associated with HG among TB-positive patients (n = 150).**

| Variable | Categories | Hypogonadism | | COR (95%CI) | AOR (95% CI) | P value |
|---|---|---|---|---|---|---|
| | | Yes n (%) | No n (%) | | | |
| **Sex** | Male | 48 (59.3%) | 33 (40.7%) | 2.6 (1.3,5.0) | 11.4 (3.6, 36.2) | <0.001* |
| | Female | 25 (36.2%) | 44 (63.8%) | 1 | | |
| **Occupation** | House wife | 14 (42.4%) | 19 (57.6%) | 1 | | |
| | Employed | 34 (57.6%) | 25 (42.4%) | 1.9 (0.8, 4.4) | 0.4 (0.1, 1.7) | 0.216 |
| | Student | 10 (43.5%) | 13 (56.5%) | 1.0 (0.4, 3.1) | 0.4 (0.1, 1.8) | 0.232 |
| | Unemployed | 10 (47.6%) | 11 (52.4%) | 1.2 (0.4, 3.7) | 0.6 (0.1, 2.9) | 0.513 |
| | Laborer | 5 (35.7%) | 9 (64.3%) | 0.8 (0.2, 2.8) | 0.2 (0.1, 0.7) | 0.051 |
| **Monthly income (ETB)** | ≤3000 (ETB) | 35 (40.7%) | 51 (59.3%) | 0.5 (0.2, 0.9) | 2.0 (0.7, 5.7) | 0.175 |
| | >3000 (ETB) | 38 (59.4%) | 26 (40.6%) | 1 | | |
| **Dietary diversity (high diet diversity)** | No | 16 (66.7%) | 8 (33.3%) | 4.1 (1.5,11.8) | 9.0 (2.4, 34.0) | 0.001* |
| | Sometimes | 17 (41.5%) | 24 (58.5%) | 1.5(0.6, 3.5) | 9.2 (2.8, 30.6) | <0.001* |
| | Usually | 25 (64.1%) | 14 (35.9%) | 3.7 (1.5, 9.1) | 3.2(1.0,10.1) | 0.042* |
| | Always | 15 (32.6%) | 31 (67.4%) | 1 | | |
| **Nausea** | Yes | 38 (55.9%) | 30 (44.1%) | 1.7 (0.9, 3.3) | 1.6 (0.7,3.6) | 0.229 |
| | No | 35 (42.7%) | 47 (57.3%) | 1 | | |
| **Treatment status** | New TB patients | 29 (54.7%) | 24 (45.3%) | 0.9 (0.3,2.6) | 1.0 (0.4, 2.5) | 0.985 |
| | On Rx ≤ 3 month | 34 (42.5%) | 46 (57.5%) | 0.5 (0.2, 1.5) | 3.6 (0.9, 14.7) | 0.76 |
| | On Rx > 3 month | 10 (58.8%) | 7 (41.2%) | 1 | | |
| **Cortisol levels (ng/ml)** | Below the mean | 26 (35.6%) | 47 (64.4%) | 1 | | |
| | Above the mean | 47 (61.0%) | 30 (39.0%) | 2.8 (1.5, 5.5) | 4.0 (1.7, 9.5) | 0.002* |

**Abbreviations:** AOR (Adjusted Odds Ratio), COR (Crude Odds Ratio), CI (Confidence Interval).

**Note:** * = Statistically Significant at P < 0.05, 1 = Reference Group, ETB = Ethiopian Birr, New = newly diagnosed TB-positive patients, RX = TB-positive patients on treatment.

biorepository (collected between 2005 and 2013). The length of storage has an impact on the accuracy of testosterone concentration measurements [49].

Male TB-positive patients had significantly higher estradiol levels compared to the TB-negative comparison group; this finding was consistent with a review article from India [46], and a study conducted in Germany [18]. This could be due to the reason that in advanced active TB, increased estradiol synthesis, driven by pro-inflammatory cytokines like interferon gamma, IL-6, and TGF-β, may have aromatase activities to convert testosterone into estradiol [21]. This is supported by the positive correlation between plasma estradiol levels, suggesting the link between cytokine activities and estradiol production [16]. This may also be augmented due to high levels of cortisol among TB patients in our study. Elevated cortisol concentration may lead to higher estradiol levels in tissue by influencing aromatase expression and androgen levels [50]. However, the current finding was contradictory to the study conducted in Ethiopia [48] and Mexico [19], which did not find a statistically significant difference in plasma estradiol levels between groups. This discrepancy might be explained by age range differences in the Ethiopian study, including younger (< 18 years old), which were dominant in the TB-negative comparison group. Since sex hormone levels vary with age, peaking after puberty [51], and with lifestyle differences with the study conducted in Mexico.

Although overall progesterone levels in male TB-positive patients did not differ significantly from the TB-negative comparison group, subgroup analysis showed that newly diagnosed TB-positive patients had significantly lower progesterone levels compared to both those on treatment TB-positive patients and the TB-negative comparison group. This finding was consistent with a study conducted in Mexico [19]. This might be explained by the increased pro-inflammatory cytokines,

specifically IL-8, produced in response to MTB antigen stimulation. Interleukin-8 negatively correlates with progesterone levels [28]. Furthermore, TB-positive patients on treatment luck a significant difference with the TB-negative comparison group; this may be due to treatment impacts, which may improve the sex hormone levels [8].

Female TB-positive patients had statistically significant lower testosterone levels compared to the TB-negative comparison group; this is contrary to the finding of the previous study conducted in India [52]. The difference could be due to differences in characteristics of study participants. The study subjects in the previous Indian study had a higher prevalence of anemia as compared to participants in our study (34 out of 75 participants), and anemia negatively impacts testosterone level [53]. The difference could also be contributed by the inclusion of women who experienced adverse events related to pregnancy (18 out of 52 pregnant women); those who were nulliparous (did not have a previous delivery) had significantly higher levels of testosterone [54]. Furthermore, TB-positive patients had significantly lower prolactin levels than the TB-negative comparison group in the previous Indian study; this may be the second reason for the discrepancy. Because prolactin was negatively associated with testosterone levels [55].

The present study revealed that there were statistically significant lower progesterone levels in TB-positive patients compared to the TB-negative comparison group, this finding aligning with previous findings conducted in Egypt [56] and India [27]. This might be explained by the increased pro-inflammatory cytokines, specifically IL-8, produced in response to MTB antigen stimulation. Interleukin-8 negatively correlates with progesterone, suggesting it may suppress ovarian sex steroid hormone secretion [28]. However, oppose the study conducted in Turkey [57], which showed no significant difference in progesterone level between groups. This may be due to study age group differences in the Turkish study; it was conducted among postmenopausal women, and as age advances, the function of ovaries diminishes [58].

In the current study, while overall estradiol levels in female TB-positive patients did not differ significantly from the TB-negative comparison group, newly diagnosed TB-positive patients had significantly higher estradiol levels in the presence of higher FSH levels compared to the TB-negative comparison group. This finding was supported by the study conducted in Turkey [57] and Nigeria [59]. During TB infection, immune cells produce such cytokines as IL-1, IL-6, TGFβ, and TNFα that activate the HPG axis. This cytokine release stimulates the production of releasing factors at the hypothalamic levels, like GnRH, leading to the pituitary synthesis of FSH and LH. Ultimately estradiol production by gonads, potentially explaining levels of elevated estradiol observed during TB infections [22,23]. However, this finding was contradictory to those conducted in India [27], Egypt [56], and Ethiopia [48]. This difference may be due to the inclusion of individuals with diabetes in the Indian and Egyptian studies; according to the reports, DM is associated with lower estradiol levels [60] and the use of stored samples in the Ethiopian study. Storage of samples for measurement of estradiol can introduce large error variance to measured concentrations [61].

The mean serum levels of male testosterone in the current study were higher than the reports obtained from Egypt [45]. This difference could be attributed to differences in participants' behavioral and clinical characteristics from the current study. Smoking status (high number of smokers, 29 out of 55 participants) and comorbidities like DM were included in the previous study. Diabetes mellitus may impair gonadal function [60,62].

The median serum testosterone levels of females in the present study were aligned with the result from India [63]. Even though the median serum values of estradiol, LH, and FSH in the present study were higher than in the study from India [63]. This difference could be attributed to methodological differences; in the previous study, median values were not calculated for males and females separately.

The prevalence of HG was significantly higher among TB-positive patients compared to the TB-negative comparison group ($p < 0.001$). The higher prevalence of HG among TB-positive patients could be due to frequent involvement of both direct [24] and indirect effects mediated by pro-inflammatory cytokines [17,19].

While few studies had investigated HG prevalence in TB-positive patients, it was higher than those reported in two Indian studies: 27% [64] and 38.1% [63]. This discrepancy may be due to differences in lifestyle factors and the fact that the previous studies may have focused solely on secondary HG.

The prevalence of HG in male TB-positive patients in this study was lower than the 73% reported in the South Africa study of PTB patients [65]; a difference may be explained by the inclusion of HIV-positive individuals in the previous study. HIV infection can directly damage the testes, causing basement membrane thickening and tubular atrophy [66].

The prevalence of HG in female TB-positive patients in the present study was in line with the study in Morocco reporting a 36.4% prevalence of premature ovarian failure among hospitalized patients [67]. Tuberculosis could affect the ovaries as EPTB, leading to reduced sex hormone production and causing HG in women [25].

Participants with a history of alcohol consumption had 4.1 times higher odds of having HG when compared to participants who did not drink [AOR = 4.13, 95% CI (1.21, 13.95)]. Alcohol consumption can affect the HPG axis loops and also can directly damage the testes in men and the ovaries in women, impairing their ability to produce sex hormones [68]. Additionally, individuals with a value of cortisol above the mean value had a significantly increased risk of developing HG, with odds being 3.4 times higher compared to those with cortisol levels below the mean value [AOR = 3.43, 95% CI (1.55, 7.46)]. Furthermore, a high cortisol value was also significantly associated with HG among TB-positive patients with the odds of 4.01. In TB, regardless of whether it's PTB or EPTB, excluding bilateral adrenal gland involvement, serum cortisol levels were elevated [69]. Prolonged hypercortisolism disrupts the HPG axis primarily by suppressing GnRH expression in the hypothalamus, leading to decreased LH and FSH secretion from the pituitary, impairing gonadal function [70,71].

This study found that male TB-positive patients had 11.36 times higher odds of HG compared to female participants [AOR = 11.36, 95% CI (3.6, 36.17)]. The higher prevalence of TB in males may correlate with a higher likelihood of developing HG in the present study.

Additionally, dietary diversity was the other factor, with participants consuming diets with lower diversity, including those with no dietary diversity [AOR = 8.98, 95% CI (2.37, 33.99)], those with sometimes [AOR = 9.2, 95% CI (2.77, 30.62)], and those with a usual dietary diversity [AOR = 3.24, (1.04, 10.06)] also having 8.98, 9.2, and 3.24 times greater odds of HG compared to those with always diverse diets, respectively. A diverse diet is generally associated with better overall health and likely supports optimal hormone production through improved nutrient intakes. However, it is not a guaranteed prevention of HG. Other factors, such as lifestyle, including alcohol consumption, smoking, stress, and underlying medical conditions, play a greater role [72].

## Strengths and limitations of the study

### Strength of the study

In contrast to most of the previous studies, this study incorporated cortisol level and employed LH and FSH, in addition to testosterone, estradiol, and progesterone, for a more comprehensive classification of HG, also considering menstrual phase in female participants for improving diagnostic accuracy. Sex hormone levels were separately analyzed by sex.

### Limitations of the study

Due to reagent constraints, hormone tests like leptin, thyroid-stimulating hormone, triiodothyronine, thyroxine, and dehydroepiandrosterone were not performed. The use of chemiluminescent immunoassay techniques other than liquid chromatography–tandem mass spectrometry (the gold standard). Latent TB was not excluded from the TB-negative comparison group due to feasibility issues. The menstrual phase was not considered for group comparison of sex hormone levels due to the cross-sectional nature of data collection. Information on diet diversity and alcohol consumption was subjectively obtained from participants and not assessed in-depth. Additionally, data on TB smear grading and type of EPTB was not identified due to missing information on patient charts.

## Conclusion

Tuberculosis infection impairs the gonadal axis and significantly alters sex hormone levels. Male TB-positive patients had significantly lower testosterone but higher estradiol, LH, and FSH, while newly diagnosed TB-positive patients had significantly lower progesterone levels compared to the TB-negative comparison group.

Female TB-positive patients had significantly lower testosterone and progesterone with higher FSH, and newly diagnosed TB-positive patients had significantly higher estradiol levels compared to the TB-negative comparison group.

The prevalence of hypogonadism was significantly higher in TB-positive patients, particularly males. In TB-positive patients, sex, dietary diversity, and cortisol levels were the significant determinant factors.

## Recommendation

**For healthcare providers.** Healthcare providers are better off include hormone (sex hormones and cortisol) analysis in TB diagnosis and management and advising patients to maintain a balanced diet.

**For tuberculosis-positive patients.** TB-positive patients should maintain a balanced diet.

**For policymakers.** policymakers should include cortisol and sex hormone analysis tests in policies to enhance early diagnosis and management of sex hormone imbalance in TB patients for better treatment outcomes.

**For funding organizations.** ought to support sex-specific, stage-specific TB studies to develop targeted therapeutic strategies.

**For future researchers.** It is better to conduct further studies with a larger sample size and matched number of participants based on sex, age, and menstrual phase using a longer follow-up study design and incorporating hormonal tests like leptin, thyroid-stimulating hormone, triiodothyronine, thyroxine, and dehydroepiandrosterone.

## Supporting information

**S1 Table. Questionnaire.**
(DOCX)

**S2 Data. This is supplementary legend that contain STROBE checklist.**
(DOCX)

**S3 Excel Data. Raw data of our manuscript.**
(XLSX)

## Acknowledgments

The authors would like to express their deep gratitude to the Department of Clinical Chemistry at the School of Biomedical and Laboratory Sciences, College of Medicine and Health Sciences, University of Gondar, for approving the ethical clearance and covering the funding resource for materials and laboratory analysis cost necessary for this study. Firstly, we are deeply grateful to the UGRCSH, Marakie, Azezzo, Poli, and Mitwab health center for their support in providing access to the study participants. Our heartfelt appreciations go to the study participants for their willingness to participate in this research. We also wish to acknowledge the clinical chemistry laboratory team for their expertise in conducting the biochemical analysis.

## Author contributions

**Conceptualization:** Eshet Gebrie.

**Data curation:** Eshet Gebrie, Abebe Birhanu, Amanuale Zayede, Elias Chane.

**Formal analysis:** Eshet Gebrie.

**Investigation:** Eshet Gebrie.

**Methodology:** Eshet Gebrie, Berihun Agegn Mengistie, Temesgen Kassie, Zeleke Kassahun, Elias Chane.

**Software:** Eshet Gebrie.

**Supervision:** Habtamu Wondifraw Baynes, Elias Chane.

**Validation:** Eshet Gebrie, Elias Chane.

**Visualization:** Habtamu Wondifraw Baynes, Temesgen Kassie, Zeleke Kassahun, Abebe Birhanu, Amanuale Zayede.

**Writing – original draft:** Eshet Gebrie.

**Writing – review & editing:** Eshet Gebrie, Elias Chane.

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
