## [Decision Letter · Decision Letter 0]

8 Sep 2025

Dear Dr. Gebrie,

Thank you for submitting your manuscript to PLOS ONE. After careful consideration, we feel that it has merit but does not fully meet PLOS ONE’s publication criteria as it currently stands. Therefore, we invite you to submit a revised version of the manuscript that addresses the points raised during the review process.

We look forward to receiving your revised manuscript.

Kind regards,

Muhammad Zubair

Academic Editor

PLOS ONE

2. In the online submission form, you indicated that [Most of data generated or analyzed during this study are included in this manuscript. Addition research data set associated with a paper is available, can be accessed up on request to the corresponding author.].

Additional Editor Comments (if provided):

Reviewer #1:

Reviewers' comments:

Reviewer's Responses to Questions

**Comments to the Author**

1. Is the manuscript technically sound, and do the data support the conclusions?

Reviewer #1: Yes

2. Has the statistical analysis been performed appropriately and rigorously?

Reviewer #1: Yes

3. Have the authors made all data underlying the findings in their manuscript fully available?

Reviewer #1: Yes

4. Is the manuscript presented in an intelligible fashion and written in standard English?

Reviewer #1: Yes

Reviewer #1: Sex hormone profile and associated factors among adult tuberculosis patients at Gondar town, northwest Ethiopia in 2024: A comparative cross-sectional study

General:

Thank you for the opportunity to review a manuscript on Sex hormone profile and associated factors among adult tuberculosis patients at Gondar town, northwest Ethiopia in 2024: A comparative cross-sectional study.

Some minor Comments

Title:

What is 2024 in the title? This is not clear to the reader, suggested to remove it in the title. The reader will find out about the year the study was done within the document.

Abstract:

Background of abstract: Is not clear and does not show the gap clearly. For example the author written that the previous reports have conflicting results, so which results (need to be clear to the reader). The author should rewrite it and clearly show the gap.

Methods of abstract: This study as specified is a comparative study and not a case control study, so advised not to use cases and control rather can say specifically as TB positive patients (or study group) and TB negative individual (comparative group). This should be changed throughout the document.

Line 33: I hope there are more than five health institutions/hospital in Gondar town, so better say “selected five hospital/health institutions”

The author need to specify the participants were matched by which characteristics.

Line 34 - 35: Specify in summery the kind of data that were collected

Line 36: State in summary how hypogonadism was defined in this study

Line 37 – 38: State the statistical software used for data analysis ( ie. STATA version…., SPSS version etc)

Results of the abstract: Indicate p-value in brackets for all results presented (p = ….)

Line 41: What do you mean by “New”. If meant newly diagnosed TB patients better write in full for the reader to be able to follow and understand

Line 44: Add the words “in female TB patients” after the word “significantly”

Conclusion and recommendation: should be grounded in the findings of your study

Line 51 – 52: Change “Early diagnose” to “Early diagnosis” and state “why early diagnosis is crucial”

Introduction:

Line 55: SPP name of the bacteria should be italicized

Line 56: Replace the first “it” with “TB”

Line 57: Change the word “damage” to “involve” , add the word “other” before the word “all”, replace “is” with the word “and”

Line 61: Add the abbreviation COVID in brackets after the long form“(COVID) and” between Reproductive health and rights

Line 65: The first sentence is not clear, rephrase

Line 76: Delete the words “sex hormone” after “pathways”

Line 83: Put full stop after “(after puberty) and then start a new sentence as “The gender bias in…………..”

Line 87: The first sentence of the paragraph lack connectivity, the author should re-write it clearly and put a reference.

Line 101 – 108: The whole paragraph is not clear, the author should re-write it to be clear to the reader.

Suggested: “Recently there has been a rise in global TB cases, deaths and spread of highly drug-resistant strains, providing alarming signals that other strategies will be needed to stop this endemic disease (29). Global control of TB can only be achieved through the concerted effort in the development of effective vaccines, improved diagnostics, as well as novel and shortened therapeutic regimens. The findings of this study will contribute to the identification of additional biomarkers that add up to the existing body of knowledge towards developing potential host markers for diagnostics, prognostics or vaccine development initiatives. Therefore, this study was aimed to assess sex hormone profile and associated factors in adult TB patients and compare with apparently healthy TB negative control.

Materials and Methods:

A lot of repetition have been observed these should be avoided.

Line 112: add “selected” before “five”

Line 114: Replace the words “used for data collection” with “involved in this study”

Line 117: Replace “by the year” with “as per year”

Study population

Line 121 – 123: The author should specify the study participants were matched by which characteristics

Line 138 – 152: Exclusion criteria: suggested to List or breakdown into slot (summarize)

Line 181 – 183: Advised to re-write the first and second sentences by combining the two in one sentence and state the kind of data collected using that questionnaire. Suggestion “The ……..data were collected using a pre-structured Amharic translated questionnaire”

Line 184: Replace “selected” with “used as”

All the data that were collected in this study and their classification should be stated under the subheading “Data collection” and not under the subheading “operational definition”. This is for the reader to easily follow and understand. Eg how BMI was classified.

The author need to state the tools used to measure weight, height and blood pressure (model, company and country of origin) eg. “Weight was measured with minimal clothing using a standard calibrated weighing scale (model, company, country of origin) Height was measured in the upright standing position using……………………….. and BMI was then calculated by the formula: weight in kg divided by height In meter squared”

Line 137: the last sentence of the paragraph is not clear, rephrase

Laboratory procedure and analysis

Considering the circadian rhythm of sex hormones, the author need to state the time of the day blood samples for hormone assay were collected

How were the blood/serum samples handled before the laboratory analysis of hormones?

Line 211: replace “separated cells from” with “obtain”, Replace “The hormonal profile” with “The sex hormone profile”

Line 212 – 214: Summarize the manufacturer in the bracket as “ (DXI800, Beckman Coulter inc, Danaher corporation company Brea, California United States)” to be placed after the word “analyzer” and then a full stop

The author also need to specify the technique used for hormone assay.

Data quality control: summarize

Statistical analysis

Line 230: Add a reference for a statistical software used for data analysis

Operational definition

Summarize and most should put under data collection part following each specific data.

Results:

Socio demographic characteristics of study participants

Line 285: change “aged” to “age”

Line 285 – 286: You stated that there were a total of 300 participants (150 TB positive and 150 TB negative controls who were sex and age matched. Why again reported to have 162 (54%) males in this study. This is confusing, the author need to clarify

Line 285 – 2888: Suggested the author to directly start reporting the two groups and their characteristics i.e “ A total of 150 TB positive patients and 150 apparently healthy negative individual as comparison group matched by sex and age were included in this study. The mean age was 32.86 ± 12.88 and 32.81 ± 12.20 for TB positive patient and a comparison group respectively (Table 1)”.

Clinical and behavioral characteristics of study participants

Line 290 – 294: Edit as examples given above for clarity

Comparison of sex hormone profile among study participants

Line 296 – 303: These 81 cases and 81 male control as well as 69 female cases and 69 female control should have been introduced early in the methodology part so that the reader can easily follow and understand not just appear over a sudden in the results section. This can be confusing to the reader, so the author is advised to revisit and do a needful amendments.

Change “cases” and “control” to “TB positive patient or study group” and “TB negative comparison group” respectively

Better to indicate specific p –value for each hormone in the brackets i.e estradiol (p = 0.0…..) , LH (p=0.0..)

Sex hormone profile in new TB cases, TB patients on treatment and controls

Line 306 -323; The narration of results is not clear to the reader, the author advised to rephrase making simple and clear to the reader.

Factors associated with alteration of sex hormone profile

Line 325 – 326: This is supposed in the first place to be narrated in the data collection part.

Line 324 – 342: Advised to change the subheading to “Comparison of sex hormone according to treatment duration among new TB positive patients” and all the results narration under these lines should be within this same subheading

The author is advised to be focused, concise and clear in results narration in the text within the results section

The author should be consistent in their table formatting. Add units where appropriate in parentheses in each table: for example, Age (years) testosterone (ng/ml). Also need to write the data with equivalent number of decimal places for all tables (i.e one decimal place for all). Also the author need to amend the title of all tables by shortening it (reducing the number of words) e. g table 1 title can be written as “Socio-demographic characteristics of study participants (N = 300)”

• For Table 1. Amend Ages (years), add mean age on the first row, shorten the title

• For Table 2. Amend cortisol (SI units), shorten title

• For table 3. Amend Testosterone (ng/ml), estradiol (SI unit),, P4 (SI unit),, LH (SI unit), FSH (SI unit), Shorten the title

• For Table 4. Amend Testosterone (ng/ml), estradiol (SI unit),, P4 (SI unit),, LH (SI unit), FSH (SI unit), Shorten the title

• For Table 5. Amend Testosterone (ng/ml), estradiol (SI unit),, P4 (SI unit),, LH (SI unit), FSH (SI unit), Shorten the title

• For Table 6, 7 and 8. Shorten the title

Cross check all the tables for data and table title page overlap and correct.

The author need to put title for all the figures.

Discussion:

Line 386: The statement “in this age group” should be rephrased and state the specific age group being referred to. Here

Line 386 – 387: Avoid repeating the objective and just directly start discussing your findings

Line 393 – 395: The advised to continue using abbreviations i.e TB, FSH, LH

Line 402: Suggested the sentence “However, opposing with the study…….” To be rephrased to “However this finding is contrary to the report of previous study in Ethiopia…………” and should be a continuation of the above paragraph.

Line 409: Add the words “and a” before the words “study conducted”

Line 410: Add the words “could be” after the word “This”

Line 413: Change the word “suggests” to “suggesting”

Line 417 - 422: This paragraph should be a continuation of the above paragraph and not a separate paragraph.

Line 419: Replace the words “methodological differences” with “differences in characteristics of study participants”

Line 423 – 425: The sentence is not clear to the reader and need to be rephrased

Line 425 - 426"This difference may be attributed to the lifestyle difference" the paragraph reiterates. How might differences in lifestyle impact the relationship between TB and progesterone levels? Simply read and provide concrete evidence.

Line 429 – 432: The sentence “Female TB patients…….” Is not clear, suggested to be rephrased as :” Female TB patients had statistically significant lower testosterone levels compared to controls. This is contrary to the finding of the previous study conducted in India (53)”. The difference could be due to difference in characteristics of study participants. The study subjects in the previous Indian study had higher prevalence of anemia as compared to participants in our study (75% vs 34% participants), and anemia negatively impacts testosterone level (54)”.

Line 432 – 436: The sentence “The difference that may be…….” be rephrased as “The difference could also be contributed by the inclusion of………….”

Line 443 – 446: The paragraph should be a continuation of the above paragraph

Line 455 – 459: The paragraph should be a continuation of the above paragraph

Line 460 – 471: The author advised to avoid too much repeat to report results in the discussion

Line 471 – 477: It is worthy to report and compare the prevalence between TB positive patients and TB negative comparison group and not the overall prevalence.

Line 513: Replacer the word “much” with “greater”

The discussion sections should be generally revised so that the paper can be reconsidered!

Strengths and limitation

Limitation of the study

Line 521 – 522: The main aim of the study to determine sex hormone profile, how comes failure to measure hormones like leptin, thyroid stimulating hormone and thyroxine considered as a limitation of this study. The author need to clarify this.

Use of techniques other than liquid chromatography-tandem mass spectrometry (Gold standard) for sex for hormone assay in this study as well as measurement of total testosterone versus free testosterone could be other limitations to be addressed. The author need to verify these.

Conclusion and Recommendation

Advised to re-write the conclusion and recommendation to be concise and clear to the reader, and both (conclusion and recommendation) should be emanating from the findings of this study.

Line 542: Add the word “analysis” after the bracket

Line 545: Add the words “analysis tests” after the word “hormones”

References

References which require editing have been observed. These need to be edited, for example Reference number 3 and 25, have some words being abbreviated or it lack some important information.

Also several grammatical error have been observed throughout the document and require to be edited in the revision.

**Do you want your identity to be public for this peer review?** For information about this choice, including consent withdrawal, please see our Privacy Policy

Reviewer #1: No

---

## [Author Response · Author response to Decision Letter 1]

17 Oct 2025

Confirmation to Continue with Former Authors and Recognition of Authorship Decision

Greetings, Editors

We appreciate your consideration of our request as well as the time and assistance you have given us during this process.

We are well aware of and respect the journal's authorship change policies. The extra authors have been eliminated in accordance with your decision and we would like to proceed with the original list of authors as previously approved.

We truly appreciate your guidance and help in handling this issue, and we look forward to continuing the publication process with your support.

---

## [Editor Report · Decision Letter 1]

23 Dec 2025

Sex hormone profiles and associated factors among adult tuberculosis patients at Gondar town, northwest Ethiopia: A comparative cross-sectional study

PONE-D-25-40268R1

Dear Dr. Gebrie,

We’re pleased to inform you that your manuscript has been judged scientifically suitable for publication and will be formally accepted for publication once it meets all outstanding technical requirements.

Kind regards,

Muhammad Zubair

Academic Editor

PLOS One
---

## [Editor Report · Acceptance letter]

PONE-D-25-40268R1

PLOS One

Dear Dr. Gebrie,

I'm pleased to inform you that your manuscript has been deemed suitable for publication in PLOS One. Congratulations! Your manuscript is now being handed over to our production team.

Kind regards,

on behalf of

Dr. Muhammad Zubair

Academic Editor

PLOS One